# Gastric Cancer: Molecular Mechanisms, Novel Targets, and Immunotherapies: From Bench to Clinical Therapeutics

**DOI:** 10.3390/cancers15205075

**Published:** 2023-10-20

**Authors:** Thais Baccili Cury Megid, Abdul Rehman Farooq, Xin Wang, Elena Elimova

**Affiliations:** Princess Margaret Cancer Centre, University Health Network, Toronto, ON M5G 2M9, Canada; thais.megid@uhn.ca (T.B.C.M.); abdulrehman.farooq@uhn.ca (A.R.F.); kevinxin.wang@uhn.ca (X.W.)

**Keywords:** novel oncogene targets, gastric cancer, immune biomarker, target therapy, immunotherapy, new therapeutic modality, translational research, clinical trials

## Abstract

**Simple Summary:**

Gastric cancer (GC) is common but often diagnosed late. Recent advances in chemotherapy, targeted therapy, and immunotherapy offer promising treatments. Perioperative chemotherapy is now the standard for resectable gastric cancer. Progress has also been made in treating metastatic disease using targeted immunotherapies. Molecular biomarkers such as PD-L1, MSI, and HER2 guide personalized treatment approaches. The review highlights these advancements and discusses future perspectives in GC treatment.

**Abstract:**

Gastric cancer is a global health concern, ranking fifth in cancer diagnoses and fourth in cancer-related deaths worldwide. Despite recent advancements in diagnosis, most cases are detected at advanced stages, resulting in poor outcomes. However, recent breakthroughs in genome analysis have identified biomarkers that hold positive clinical significance for GC treatment. These biomarkers and classifications offer the potential for more precise diagnostic and therapeutic approaches for GC patients. In this review, we explore the classification and molecular pathways in this disease, highlighting potential biomarkers that have emerged in recent studies including targeted therapies and immunotherapies. These advancements provide a promising direction for improving the management of GC.

## 1. Introduction

Gastric cancer (GC) is a significant global health concern, ranking as the fifth most common cancer and the fourth leading cause of cancer-related deaths worldwide in 2020, according to the Global Cancer Observatory (GLOBOCAN) [1]. Its incidence is particularly pronounced in East Asia and Eastern Europe [1]. More than 95% of GC cases are adenocarcinomas. Detecting early GC poses a significant challenge due to the delayed emergence of clinical symptoms, which precludes surgical intervention. Considering poor survival rates among GC patients worldwide, especially those with metastatic disease, there is a tremendous unmet need to uncover novel therapeutic approaches to improve patient outcomes [2].

The mechanisms behind GC pathogenesis remain unclear and involve a complex interplay of environmental and genetic factors [2]. Dysregulation of numerous genes and pathways plays a pivotal role during gastric carcinogenesis. The recent understanding of molecular signaling pathways is contributing to our knowledge of the tumor’s pathogenesis and provides valuable insights into the potential of targeted therapies [3]. Consequently, new molecular classifications have been proposed to better understand GC and its subtypes [4,5].

In this detailed review, our goal is to review the complex biological processes and molecular mechanisms involved in GC and its carcinogenesis. We focused on the different molecular subtypes of GC and their possible uses in practical applications and summarizing the latest approved therapies.

## 2. Histological and Molecular Classification

Various classification systems exist for gastric cancer as depicted in Figure 1 [6]. The Lauren classification, established in 1965, divides it into intestinal and diffuse types based on histopathology [6]. The 2010 World Health Organization (WHO) classification, revised in 2019, organizes gastric cancer into distinct subtypes of adenocarcinomas, including papillary, tubular, mucinous, poorly cohesive, and less common forms of gastric tumors [7].

There was a need for a new molecular classification system since histological classifications do not take into account the molecular profiling features in gastric cancer [4]. The Cancer Genome Atlas (TCGA) [4] proposed a classification that identifies dysregulated pathways and potential gene mutations in four distinct molecular subtype groups. The first subtype comprises Epstein–Barr virus (EBV)-positive tumors (9%) characterized by high levels of DNA hypermethylation. The second subtype includes microsatellite instability (MSI) tumors (22%) with a hypermutated genome, DNA hypermethylation, and MLH1 silencing. The third subtype consists of genomically stable (GS) tumors (20%) exhibiting a low mutation burden and displaying more aggressive disease traits. Over 70% of these tumors demonstrate diffuse histology and harbor somatic CDH1 mutations and Claudin 18.2 rearrangements. The final subtype is chromosomal instability (CIN) tumors (50%) which are associated with intestinal histology and exhibit high somatic copy number aberrations, common TP53 mutations, and amplifications of the RAS receptor tyrosine kinase pathway. In the CIN subtype, most affected genes are vascular endothelial growth factor (VEGF), epidermal growth factor receptor (EGFR) (10%), Human epidermal growth factor receptor (ERBB2) (24%), ERBB3 (8%), FGFR2 (8%), and c-Met (8%) (Figure 2).

The Asian Cancer Research Group (ACRG) [5] conducted a study that built on the TCGA analysis, incorporating additional molecular analysis, and correlating the results with clinical outcomes data. They identified four distinct subtypes of gastric cancer: MSI-High (23% of patients) with the best prognosis, typically diagnosed in early-stage cancer; microsatellite stable (MSS) and epithelial–mesenchymal transition (EMT) (15% of patients) with the poorest overall prognosis, associated with diffuse histology (80%) and advanced clinical stages (III/IV); MSS/TP53 intact (26% of patients) with the second-best prognosis and a high rate of EBV infection (15% of patients) and finally, MSS/TP53 loss (36% of patients) with the highest rate of TP53 mutations and a less favorable prognosis than MSI and MSS/epithelial/TP53 intact, but better than MSS/EMT [5].

Figure 1 illustrates the major features and genomic alterations associated with each GC subtype according to Histological (Lauren and WHO) and Molecular (TCGA and ACRG studies) classification.

## 3. Risk Factors and Molecular Mechanisms

Hereditary syndromes and familial GC, characterized by specific family history criteria, occur in less than 10% of cases [8]. Ninety percent of GC is non-hereditary and associated with several factors, occurring sporadically. Non-cardia tumors are often associated with *Helicobacter pylori* infection [9], with chronic infection leading to chronic gastritis, and continued inflammation leading to damage to the stomach lining, ulcers, and gastric atrophy [10]. *H. pylori* inflammation is primarily caused by pathogenic factors like cytotoxin-related gene A (CagA), vacuolar cytotoxin A (VacA), and the Cag IV secretion system, encoded by specific genes within the CagA pathogenicity island (PAI) DNA insertion element [11]. These virulence factors are released, activating signaling pathways such as NF-κB, p53 and JAK-STAT. A detailed discussion of these pathways will be discussed in Section 4. The TP53 gene exhibited increased activity in individuals with H. *pylori* infection, as evidenced by both the TCGA database and its RNA sequencing data [4]. A total of 33 signaling pathways and 10 biological processes demonstrated a significant positive correlation with *H. pylori* infection [4].

Both cardia and non-cardia tumors share risk factors such as alcohol consumption and tobacco smoking [12]. Tumors situated in the cardia region are, on the other hand, more likely to be linked with gastroesophageal reflux and EBV infection [13]. EBV infection may lead to increased DNA hypermethylation, frequent PIK3CA mutations, and overexpression of JAK2 [13]. Furthermore, programmed cell death protein ligand 1 (PD-L1), expressed in certain tumor cells, inhibits immune responses by binding to PD-1 receptors on T lymphocytes. Elevated levels of PD-L1, whether on tumor cells or infiltrating lymphocytes, correlate with improved response rates and overall survival (OS) when treated with immunotherapy in different tumors [14]. A systematic review estimated a PD-L1 positivity rate of approximately 55% in EBV-associated gastric cancer, but with significant variability among studies, likely linked to histologic variations. Specifically, GC-LS (lymphoid stroma) tumors, often associated with EBV, exhibit a PD-L1 positivity rate exceeding 80%. In EBV-GC, higher PD-L1 expression is observed, suggesting the potential of EBV positivity as a marker for immunotherapy selection [14].

The key molecular mechanisms involved in gastric cancer development caused by EBV will be discussed further. See discussions in Section 4.

## 4. Molecular Mechanisms and Signaling Pathways in Gastric Cancer

In 2014, the TCGA Research Network conducted a groundbreaking study that represented one of the most comprehensive molecular characterizations of gastric adenocarcinoma [4]. The study involved the evaluation of 295 primary gastric adenocarcinomas, employing a comprehensive approach encompassing six molecular platforms: array-based somatic copy number analysis, whole-exome sequencing, array-based DNA methylation profiling, messenger ribonucleic acid (RNA) sequencing, microRNA (miRNA) sequencing, and reverse-phase protein array (RPPA). Additionally, MSI testing was performed on all tumors, adding a crucial dimension to the molecular analysis. Through this rigorous examination, TCGA was able to categorize gastric cancer into four distinct subtypes, its unique pathways and genomic features involved in gastric adenocarcinoma (Figure 2).

However, the tumorigenesis process entails a diverse array of signaling pathways, beyond those spotlighted in Figure 2, as depicted in Figure 3. Within the context of gastric cancer, we delve into these specific signaling pathways that influence invasion and tumor progression. Furthermore, the subsequent section will thoroughly explore signaling pathways, their frequencies, their prognostic implications, and their significance in both gastric cancer research and treatment.

### 4.1. MAPK Signaling Pathway

The mitogen-activated protein kinase (MAPK) signaling pathway represents an extensive group of serine/threonine protein kinases crucial for determining how cells respond to various external triggers [14]. In GC, the RAS/RAF/MAPK and PI3K/AKT/mTOR signaling pathways are notable among the highly impaired pathways [15].

The MAPK signaling pathway is composed of five interconnected cascades: extracellular signal-related kinases (ERK), specifically ERK1/2; Jun amino-terminal kinases (JNK), encompassing SAPK/JNK1, JNK2, and JNK; and p38-MAPK, which involves p38α, p38β, p38γ, and p38δ, along with ERK5 and ERK3 [16]. In GC, the MAPK signaling pathway is responsible for the processes of invasion and metastasis, such processes involve motility and cell adhesion and disintegration of focal adhesions triggered by the epidermal growth factor receptor (EGFR), which is regulated by activating the ERK/MAPK pathways [16].

The JNK subgroup within the MAPKs family responds to stress signals, a trigger for cell proliferation, apoptosis, or transformation [17]. A study involving mice highlighted JNK’s association with tumor initiation and progression, suggesting it as a promising target for GC prevention [18].

Within cancer, the p38-MAPK pathway exhibits intricate regulation. Numerous studies have indicated that p38 serves as an oncogenic factor, playing a pivotal role in pathological events linked to tumor progression, such as inflammation, invasion, angiogenesis, and chemotherapy resistance in GC [19].

Moreover, miRNAs, like miR-302b, MiR-20a, miR-21, and miR-106a possess regulatory abilities in various cellular pathways. These miRNAs have been recognized as controllers of MAPKs, modulating the proliferation, survival, and metastasis of gastric cancer cells [20]. Through experiments conducted on GC cell lines, it was observed that miR-491-5p effectively suppressed cell migration and proliferation while promoting apoptosis. Initially identified as an inhibitor of the antiapoptotic factor BCL-XL, miR-491-5p has been demonstrated to inhibit ERK1/2 and AKT signaling pathways [20].

### 4.2. HER2 Signaling Pathway

Human epidermal growth factor receptor 2 (HER2 or erb2) is an oncogene encoded by erb2 on chromosome 17 and belongs to the EGFR family of receptor tyrosine kinases [21]. In HER2-induced cancers, particularly HER2-positive tumors, the HER2/HER3 dimer plays a crucial role in tumorigenesis and tumor maintenance as the most potent heterodimer within the EGFR family. Hence, directing efforts towards inhibiting the association of HER2 with other EGFR family counterparts, especially HER3, represents a viable therapeutic strategy for addressing HER2-positive tumors [22].

The oncogenic Impacts of elevated HER2 levels arise from the spontaneous formation of receptor homodimers or heterodimers with other members of the EGFR family. This activation initiates downstream signaling cascades, including the PI3K/AKT/mTOR and MAPK/ERK1/2 pathways. The activation of these pathways fosters diverse cellular processes vital for tumor progression, such as cell proliferation, differentiation, survival, angiogenesis, and metastasis [21,22].

The phase III Trastuzumab for Gastric Cancer (ToGA) trial reported the incidence of HER2-positive gastric cancer to be 22% [23], but it can vary in the literature. The PI3K/AKT/mTOR signaling pathway is implicated in the molecular mechanisms associated with HER2-driven tumorigenesis [24]. Therefore, targeting HER2 and its downstream signaling pathways holds potential as a therapeutic strategy for HER2-positive tumors [21].

### 4.3. PI3K/AKT/mTOR Signaling Pathway

The PI3K/AKT/mTOR pathway is frequently dysregulated in GC [25]. PIK3CA mutations are present in approximately 9–12% patients with non-hypermutated tumors and 32% patients with hypermutated tumors [25,26]. A range of mutations within the PIK3CA gene and amplification of receptor tyrosine kinase genes such as EGFR and HER2 were detected in the analyzed cases of gastric cancer [27]. While some studies have not found a significant association between PIK3CA mutations and clinical outcomes, the amplification of PIK3CA is closely linked to the advancement of tumors, prognostic outcomes, and the emergence of drug resistance in GC [27].

In GC, the PI3K/AKT/mTOR signaling pathway assumes a pivotal role, propelling tumor progression via a diverse range of mechanisms such as impeding apoptosis, fostering drug resistance, facilitating metastasis, and promoting angiogenesis [28]. Perturbations in this pathway notably play a key part in resistance to HER2-targeted therapies and the development of chemoresistance in breast cancer [29]. Given the substantial engagement of the PI3K/AKT/mTOR signaling pathway in the progression of GC, aiming interventions at this intricate signaling axis represents a promising yet challenging therapeutic strategy for effective GC treatment.

### 4.4. HGF/c-MET Signaling Pathway

The mesenchymal epidermal transition factor (c-MET), which is encoded by the proto-oncogene MET, is a transmembrane receptor expressed on the surface of epithelial and endothelial cells [30]. c-MET is a receptor tyrosine kinase belonging to the MET (MNNG HOS transforming gene) proteins. Its specific ligand is hepatocyte growth factor (HGF). When HGF binds to c-MET, it activates the canonical pathway by causing c-MET to form homodimers and phosphorylate its intracellular kinase domains [30]. The HGF/c-MET pathway plays crucial roles in normal cellular processes, but its abnormal activation is strongly linked to tumor invasion and metastasis in various epithelial cancers. This aberrant activation can occur through multiple mechanisms including gene amplification, activating mutations, changes in gene expression, overexpression of c-MET or HGF, increased stimulation by self-produced or neighboring HGF molecules, the interaction with other active receptors on the cell surface, and disruptions caused by environmental factors such as low oxygen levels and inflammation [31]. While gene amplification of MET is infrequent in gastric cancer p”Iie’ts (4–10%) [32], an overexpression of the c-MET protein has been identified in a significant proportion of cases, ranging from 18% to 82% [33]. The complexity of the MET signaling pathway, the lack of consensus and poor biomarker determination as well as the diverse resistance mechanisms (crosstalk, novel bypass mutations, upregulated gene amplification), resulted in the limitation of clinical efficacy of MET inhibition [34].

### 4.5. Wnt/β-Catenin Signaling Pathway

The Wnt/β-catenin signaling pathway is involved in cell proliferation, migration, and cell death, and is important for the development and homeostasis of various tissues [35]. Abnormal activation of the Wnt/β-catenin pathway, crucial in the malignant transformation and invasion of GC, is brought about by mutations in various key components of the standard Wnt signaling [36]. This pathway becomes activated when Wnt ligands connect with Frizzled (FZD) and LRP5/6 receptors, initiating the recruitment of intracellular proteins like Disheveled (DVL) and Axin. This activation inhibits the phosphorylation of β-catenin and ensures its stability. Accumulation of β-catenin within the cytoplasm permits its dissociation from degradation complexes, facilitating its entry into the nucleus. Inside the nucleus, β-catenin collaborates with T-cell factor/lymphoid enhancer factor (TCF/LEF) transcription factors to activate genes responsive to Wnt signaling [37]. Disrupted activation of the Wnt/β-catenin pathway, frequently resulting from mutations in its constituents, plays a role in the malignant transformation and invasion of GC [38].

Furthermore, this pathway plays a role in governing the infiltration of T cells within the tumor microenvironment and influences the response to programmed cell death protein 1 (PD-1) antibodies. Additionally, activated Wnt/β-catenin signaling in GC brings about further mechanisms, such as the adjustment of immunoregulatory factors like CCL28 (Mucosae-associated epithelial chemokine) and the interplay between β-catenin and E-cadherin [39]. An intriguing finding is the unexpected antitumor effect observed upon CCL28 blockade, which effectively hinders the infiltration of Treg cells. This discovery holds significant importance as it introduces a novel concept for the immunotherapy of gastric cancer, potentially leading to innovative diagnostic approaches and therapeutic interventions in the foreseeable future [40].

### 4.6. FGF/FGFR Signaling Pathway

The fibroblast growth factor receptors (FGFRs) represent a group of transmembrane proteins found across various cell types [41]. This family encompasses four distinct members, including FGFR1, FGFR2, FGFR3, and FGFR4. In the context of GC, common FGFR alterations include mutations in FGFR1, amplification of FGFR2, and rearrangements in FGFR3 [41]. The activation of FGFRs occurs upon binding of fibroblast growth factors (FGFs), leading to the phosphorylation of the intracellular tyrosine kinase domain and the initiation of several critical cellular pathways [42]. These pathways encompass the RAS/MAPK, PIK3CA/AKT/mTOR, and Janus kinase (JAK) pathways, which can influence various processes like angiogenesis, cell mitosis, differentiation, proliferation, and invasion [41,42].

The prevalence of FGFR abnormalities in GC ranges from 4.1% to 7% [43,44], with amplifications being the most frequently observed, followed by rearrangements and mutations. Cancers exhibiting co-alterations in FGFR2, c-Jun, and YAP1 have shown associations with poorer clinical outcomes [45]. Moreover, FGFR has emerged as a potential prognostic biomarker and therapeutic target in inhibiting the development of gastric tumors [46,47,48].

### 4.7. HIF-1α Signaling Pathway

HIF-1α plays a crucial role in cellular adaptation to low oxygen levels. When cells experience hypoxia, HIF-1α expression increases, and the activity of hydroxylases, which normally inhibit HIF-1α, is suppressed due to the absence of oxygen. This leads to the activation of HIF-1α, which then moves to the cell nucleus. Once in the nucleus, HIF-1α functions as a transcription factor, influencing the regulation of various target genes involved in metabolism, inflammation, vascular homeostasis, and tumor formation [49]. The HIF-1α signaling pathway is believed to play a significant role in advancing the progression of GC by facilitating tumor cell growth, promoting angiogenesis, inducing EMT, fostering resistance to therapy, and inhibiting programmed cell death. Increased expression of HIF-1α in GC patients may serve as an indicator of unfavorable OS outcomes [50].

### 4.8. Claudin 18.2 Signaling Pathway

Claudins are transmembrane proteins which maintain the tight junction between cells which form a paracellular barrier to control the flow of molecules between cells. Claudin-18 (CLDN18.2) has been linked with prognosis in GC [51]. Aberrant expression of CLDN18.2 is commonly observed in the initiation and progression of various malignancies. In the context of gastric epithelial tissue undergoing malignant transformation, disruptions in cellular polarity result in the exposure of CLDN18.2 epitopes on the cell surface, leading to its highly selective and stable expression in specific tumor tissues [52]. The CLDN18.2 protein plays a role in tumor cell proliferation, differentiation, and migration. The stomach-specific isoform, CLDN18 isoform 2 (CLDN18.2) [53], is emerging as a promising treatment target because of high expression in GC cells, including targeting via adoptive T-cell strategies. According to a meta-analysis the expression of CLDN18.2 was detected in approximately 34.2% of a combined population of 2055 patients [54].

### 4.9. NF-κB Signaling Pathway

The NF-κB family of transcription factors, including RelA, RelB, c-Rel, NF-κB1 (p50), and NF-κB2 (p52), regulate the expression of various genes involved in cell survival, apoptosis, and inflammation [55]. Activation of the canonical NF-κB pathway occurs when receptors such as TNFRs, TLRs, and IL-1R stimulate the IκB kinase (IKK) complex, leading to the degradation of IκBα. This allows NF-κB to translocate to the nucleus and activate downstream target genes, contributing to GC progression and metastasis [56]. Aberrant NF-κB signaling is associated with anti-apoptotic factor in GC [57]. The inhibition of NF-κB signaling shows challenging potential in inducing apoptosis, cell cycle arrest, and enhancing the efficacy of chemotherapy. The PI3K/AKT pathway regulates the NF-kB cascade, leading to NF-kB activation promoting cell invasion and migration [57].

### 4.10. TGF-β Signaling Pathway

Transforming growth piter-β (TGF-β) is an f of polypeptides involved in various physiological processes, including embryonic growth and inflammation regulation [58]. TGF-β1, the most abundant form, has complex roles in cell growth regulation and is linked to tumor development. In GC, TGF-β1 influences clinical features and patient prognosis, sometimes inhibiting cell growth but also promoting tumor progression. TGF-β signaling induces an epithelial–mesenchymal transition (EMT) in GC through the interaction with the AMPK pathway. Targeting Smad3 phosphorylation in TGF-β signaling may offer a potential therapeutic strategy for GC [58,59,60].

### 4.11. P53 Signaling Pathway

P53 holds a central role in overseeing DNA repair and governing the cell cycle, apoptosis, and differentiation by means of interactions between DNA and proteins, as well as interactions between proteins themselves [61]. It holds significant importance in averting the buildup of potentially cancerous or flawed cells by inducing aging, promoting cell apoptosis, and facilitating DNA repair. Elevated expression levels of p53 have been observed in GC patients, with a TP53 gene mutation rate of approximately 30% across all GC patients. However, this mutation rate may vary among patients with different GC subtypes and etiologies [62]. The presence of *H. pylori* infection has been linked to the accumulation of TP53 gene mutations, reported in approximately 50% of gastric tumors [63]. This occurs due to genotoxic stress, where p53 activates signaling pathways that result in temporary cell cycle arrest, allowing for DNA repair processes to take place. Inactivation of p53 promotes genomic instability, a characteristic feature of cancer, further underscoring its significance in GC development [63].

### 4.12. STAT3 Signaling Pathway

Signal transducers and activators of transcription 3 (STAT3) is a hyperactivated oncogene found in various cancers, including GC. It is activated by cytokine binding to transmembrane receptors, leading to dimerization and transphosphorylation of JAKs [64]. Upon activation, STAT3 relocates to the nucleus, where it modulates the expression of genes linked to the proliferation, invasion, and resistance to chemotherapy in cancer cells [65]. Both STAT3 and Survivin hold promise as potential indicators for prognosis and targets for therapeutic interventions in GC [66]. The JAK2/STAT3 pathway is also implicated in GC metastasis and epithelial–mesenchymal transition. MicroRNAs, long non-coding RNAs, and circular RNAs have been identified as regulators of STAT3 in GC, potentially influencing its expression levels and contributing to chemoresistance and tumor progression [67].

### 4.13. PDL-1/PD-1/CTLA-4

Cancer growth and progression can also be related to the suppression of the immune system. Immune checkpoints assume a pivotal function in this process [68]. These checkpoints can either stimulate or inhibit immune cell activity, maintaining a balance in the immune response. Some inhibitory checkpoints like PD-1, CTLA-4, TIM-3, LAG-3, and TIGIT are expressed on T cells and regulate immune signaling pathways, preventing excessive immune damage [69]. In tumor cells, these checkpoints are upregulated during tumor progression, suppressing the body’s ability to mount an effective anti-tumor immune response, and allowing the tumor to evade immune attack. Therefore, targeting immune checkpoints has become a vital approach in cancer immunotherapy [68].

One of the well-studied pathways is the PD-1/PD-L1 signaling, where PD-L1 and PD-L2 proteins on cancer cells interact with PD-1 on T cells, reducing T cell activity and promoting cancer cell survival [69]. Similarly, the CD28/CTLA-4/B7 pathway involves proteins like CD28 and CTLA-4 on T cells and B7-1/2 on antigen-presenting cells. The binding of B7-1/2 to CTLA-4 inhibits T-cell activity, preventing the killing of tumor cells [69]. Blocking these interactions using immune checkpoint inhibitors, like anti-CTLA-4 antibodies, can activate T cells and enhance their ability to target and eliminate cancer cells [69]. Other pathways involving TIM-3, LAG-3, and TIGIT also contribute to immune tolerance and dysfunction, making them potential targets for cancer therapy as well [70]. Among these, PD-1/PD-L1 signaling has received extensive attention as a diagnostic and prognostic biomarker and a therapeutic target for GC treatment.

### 4.14. MSI High

In normal cells, the DNA mismatch repair (MMR) system ensures genomic accuracy during DNA replication by recognizing and repairing genetic mismatches. However, in MSI tumor cells with deficient MMR (dMMR), microsatellite DNA mismatches accumulate, leading to mutations in various genomic regions, primarily involving key MMR components like MLH1, MSH2, MSH6, and PMS2/1 [71]. Disruption of MMR proteins can occur due to mutations within the coding region, promoter methylation, or chromosomal rearrangements, ultimately resulting in loss of heterozygosity.

In gastric cancer, MSI prevalence varies with tumor stage, being highest in node-negative disease (up to 20%) and lowest in metastatic disease (<5%), with a higher occurrence in the intestinal type [72]. While hereditary Lynch syndrome is a rare cause (~1.5%) of GC, epigenetic silencing of hMLH1 through promoter hypermethylation is a leading factor in MMR deficiency for both sporadic and familial MSI GCs [73].

Extensive research has advanced our understanding of the molecular landscape of MSI GCs. The whole genome sequencing performed by the TCGA group has elucidated genes significantly mutated in MSI GCs, encompassing crucial processes like cell cycle progression and regulation (e.g., TP53, IGFIIR, and TCF4), DNA integrity maintenance (e.g., hMSH6, hMSH3, MED1, RAD50, BLM, ATR, and MRE11), chromatin remodeling, cell death (e.g., RIZ, BAX, CASPASE5, FAS, BCL10, and APAF1), transcription regulation, and signal transduction [4]. Additionally, mutations affecting major histocompatibility complex class I genes, such as B2M and HLA-B, contribute to a loss of HLA class 1 complex expression, reducing antigen presentation and enabling an immune-surveillance escape.

Moreover, specific mutations have been identified in genes like EGFR, KRAS, PIK3CA, and MLK3, with varying prevalence across MSI cases [74]. The association between KRAS mutations and MSI status has been firmly supported by numerous studies (30% in MSI tumors) [74]. Other genes frequently mutated in MSI GC, such as ARID1A and RNF43, further contribute to the intricate molecular landscape.

## 5. Novel Targeted Treatments in Gastric Cancer

### 5.1. HER-2-Directed Therapy

Human epidermal growth factor receptor 2 (HER2), also known as ERBB2, is a part of the epidermal growth factor receptor (EGFR) family, and its overexpression/amplification is a driver of tumorigenesis in GC [75]. The phase III ToGA (Trastuzumab for Gastric Cancer) trial involved the addition of trastuzumab, a monoclonal antibody which induces antibody-dependent cellular cytotoxicity (ADCC) by binding to the extracellular domain (domain IV) of HER2, to standard chemotherapy in first-line treatment of HER2-positive advanced gastric or gastroesophageal junction cancer [23]. It demonstrated improvement in median OS of 13.8 months for the chemotherapy and trastuzumab group as compared to 11.1 months for those receiving chemotherapy alone. It also showed a greater benefit for patients with an immunohistochemistry score for HER2 at 3+, or IHC 2+ and Fluorescence in situ hybridization (FISH) positive, in whom OS was 16 months for the chemotherapy and trastuzumab group as compared to 11.8 months to those receiving chemotherapy alone [23]. The trial established the addition of trastuzumab to standard chemotherapy as a standard of care for HER2-positive advanced gastroesophageal and gastric adenocarcinoma.

Not all HER2-directed agents have been found successful in GC treatments [76]. Pertuzumab is a humanized monoclonal antibody that targets the heterodimerization domain of HER2 inhibiting HER2 heterodimerization which prevents downstream signaling [77]. It was used successfully in combination with trastuzumab and chemotherapy in the CLEOPATRA study group, a phase III trial in the first-line setting in metastatic HER2-positive breast cancer [77]. The JACOB study trial (NCT01774786) similarly involved the addition of pertuzumab or placebo in combination with chemotherapy and trastuzumab in HER 2-positive metastatic gastric and gastroesophageal junction cancer but did not show improved OS [78]. Lapatinib, a small-molecule tyrosine kinase inhibitor targeting HER2 and EGFR, and approved for the management of HER2-positive metastatic breast cancer, was tested in the GC population as part of the phase III TyTAN (Lapatinib Plus Paclitaxel Versus Paclitaxel Alone in the Second-Line Treatment of HER2-Amplified Advanced Gastric Cancer in Asian Populations) and LOGiC (Lapatinib in Combination With Capecitabine Plus Oxaliplatin in HER2-Positive Advanced or Metastatic Gastric, Esophageal, or gastroesophageal adenocarcinoma) trials, with both showing negative results [79,80].

In addition to monoclonal antibodies such as trastuzumab and pertuzumab, there has been the development of bispecific HER2-directed antibodies. Zanidatamab or ZW25 is a novel, humanized, bispecific, immunoglobulin G isotype 1-like, monoclonal antibody directed against both the juxta membrane extracellular domain (domain IV) and the dimerization domain (domain II) of HER2 [81]. After adequate tolerability and antitumor activity were demonstrated in a phase I study, it has shown encouraging overall response rates and disease control rates when used in combination with standard chemotherapy in HER2-positive gastroesophageal adenocarcinoma [81,82]. Results from a phase II first-line trial of a combination of zanidatamab with standard chemotherapy show an impressive objective response rate of 79%, a 12-month survival rate of 88% and a median OS that has not been reached [83]. KN026 and PRS-343 are other bispecific antibodies in development [84,85].

There has also been recent development of HER2-directed antibody–drug conjugates (ADC). Trastuzumab emtansine (TDM-1) is an ADC of trastuzumab conjugated to maytansine, an antitubulin agent [86]. The GATSBY phase II/III trial compared TDM-1 with a taxane in the second-line treatment of metastatic HER2-positive gastric or gastroesophageal cancer but found no improvement in outcomes [87]. Trastuzumab Deruxtecan (TDxd) is another ADC of trastuzumab conjugated with a DNA topoisomerase I inhibitor via a tetrapiptide cleavable linker [88]. The phase II Destiny-Gastric-01 trial compared trastuzumab–deruxtecan with standard chemotherapy in patients who received at least two previous therapies in the advanced HER2-positive gastric cancer and showed significant improvement in response rate and OS [89]. The high drug-to-antibody ratio and potential for bystander effect, make the case for its potential use in HER2-low gastric and gastroesophageal cancers [90]. An exploratory analysis from the Destiny-Gastric 01 trial confirmed activity for TDxd in patients with HER2-low (immunohistochemistry 2+/in situ hybridization–negative; immunohistochemistry 1+) gastric/gastroesophageal adenocarcinoma who had received at least two prior lines of treatment but were HER 2-naïve [91]. It is important to note that there is a known association with TDxd and the development of interstitial lung disease or pneumonitis, with 10.5% of patients receiving TDxd being affected in the trial of TDxd against TDM-1, and this was significantly greater than rates of pneumonitis seen with TDM-1 at 1.9% [88]. These levels of toxicity are significant, particularly for a targeted agent such as this. More robust evidence from randomized control trials is awaited.

The DESTINY-Gastric02 multicenter study involved 79 adult patients with unresectable or metastatic gastric or gastroesophageal junction cancer who had progressive disease on or after first-line therapy with a trastuzumab-containing regimen [92]. They were treated with trastuzumab deruxtecan, and the primary endpoint was the objective response rate. In the primary analysis, 38% of patients achieved a confirmed objective response, with 4% experiencing complete responses and 34% partial responses.

It is important to note that the most common resistance mechanism to HER2-directed therapy is loss of HER2 expression [91]. Studying paired samples of patients prior to receiving HER2-directed therapy and at the time of progression confirms the observation of loss of HER2 positivity and HER2 overexpression [91].

CAR-T cells targeting HER2 have shown promising results in treating solid tumors [93]. The CAR design includes an extracellular antigen-binding region, a transmembrane region for signal transduction, and an intracellular signal transduction region. CAR-T cells can enhance the immune response by releasing factors that kill tumor cells. However, before widespread clinical use for GC more large-scale, high-quality trials are needed to confirm its efficacy and safety.

### 5.2. FGFR Agents

This pathway has been targeted in GC most notably with bemarituzumab, a recombinant, humanized, IgG1 kappa monoclonal antibody that binds to the extracellular domain of FGFR2b, inhibits FGFR2B activation, and causes enhanced ADCC against cells harboring FGFR2b [47]. In the phase II FIGHT trial, patients with HER2-negative, FGFR2b-selected advanced gastric or gastroesophageal junction adenocarcinomas in the first-line setting were randomized to receive 5-fluorouracil, leucovorin, and oxaliplatin chemotherapy (mFOLFOX6) in combination with bemarituzumab or placebo [47]. Eligibility was defined as FGFR2b overexpression using immunohistochemistry and plasma next-generation sequencing of cell-free circulating tumor DNA (ctDNA) for FGFR2 amplification. The study met its primary end point with significantly improved progression-free survival (PFS) (9.5 versus 7.4 months, HR: 0.68, 95% CI: 0.44–1.04) and OS (19.2 versus 13.5 months, HR: 0.60, 95% CI: 0.38–0.94) [49]. It also hinted at an association between FGFR2b expression and response, as patients with an FGFR2b expression of more than 10% on immunohistochemistry had an OS of 25.4 months versus 11.1 months (HR: 0.41, 95% CI: 0.23–0.74). It is important to note that not all attempts to target FGFR in GC have been successful. AZD4547, a selective FGFR1/2/3 TKI with potent preclinical activity in FGFR2-amplified GC patient-derived xenograft model, has been evaluated in the phase II trial in the second line, with no improvement in the primary end point of PFS [48].

The FORTITUDE-101 is an ongoing phase III study investigating the efficacy and safety of bemarituzumab in combination with mFOLFOX6 chemotherapy versus placebo and mFOLFOX6 in untreated, unresectable advanced gastric or GEJ adenocarcinoma patients with FGFR2b overexpression [94]. Approximately 516 patients will be enrolled, and the primary endpoint is OS, with secondary endpoints including PFS, objective response, and adverse events.

### 5.3. VEGF Agents

Angiogenesis has been noted to be a hallmark of cancer and is, therefore, a therapeutic target in GC, as in many other cancers. The VEGF family, consisting of three receptors, VEGFR1–3, and six growth factors, VEGFA–E, are key regulators of tumor angiogenesis, and have been targeted by various anticancer agents [95]. Ramucirumab is a fully humanized monoclonal antibody, directed against VEGFR2 [95,96]. The phase III REGARD trial evaluated ramucirumab as compared to placebo in the second-line setting in gastric or gastroesophageal junction (GEJ) cancers and showed an improved median OS of 5.2 months compared with 3.8 months for placebo [96]. The phase III RAINBOW trial compared Ramucirumab and paclitaxel versus placebo and paclitaxel in the second-line setting in advanced gastric or gastroesophageal junction adenocarcinoma and found a significantly longer median OS of 9.6 months for the ramucirimab/paclitaxel arm as compared to versus 7.4 months for the placebo/paclitaxel arm [97].

Regorafenib is a small molecule inhibitor targeting multiple intracellular tyrosine kinases which includes angiogenesis-related pathways (VEGFR1 and 2), stromal factors (PDGFRβ), and oncogenic drivers (BRAF, RET, KIT) [98]. The phase II INTEGRATE trial involved the randomization of patients to receive regorafenib or placebo, in a population of previously treated advanced GC (esophagogastric junction or stomach, adenocarcinoma, or undifferentiated histology) [98]. It showed an improved PFS of 2.6 months for the regorafenib arm as compared to 0.9 months for placebo [98]. The INTEGRATE IIb- trial is an ongoing open-label, randomized trial comparing a combination of regorafenib and nivolumab with standard chemotherapy in pre-treated patients with advanced gastroesophageal cancers, and results of this are awaited [99]. Apatinib is another small-molecule TKI that selectively binds to and inhibits VEGFR-2 [100]. A randomized phase III placebo-controlled trial conducted in Asia in patients with gastric and gastroesophageal junction tumors who had progressed on two or more prior lines of treatment showed a significant improvement in OS in the apatinib group with a median OS of 6.5 months against 4.7 months for the placebo arm [100].

### 5.4. NTRK Agents

Tropomyosin receptor kinase (TRK) proteins TRKA, TRKB, and TRKC, are a part of the transmembrane tyrosine kinases [101]. Several cancers in children and adults have been shown to harbor chromosomal TRK fusions which lead to overexpression of the chimeric protein, leading to oncogene addiction [101]. They are exceedingly rare in solid tumors; however, with a prevalence of 1% across tumor types [102]. This pathway has been targeted with the introduction of larotrectinib and entrectinib, both of which are first-generation pan-TRK inhibitors with activity against TRKA, TRKB, and TRKC. Larotrectinib, was evaluated as part of a study, with patients prospectively identified to have TRK fusion-positive tumors and showed a marked and durable antitumor response of 75% by the independent review. Entrectinib similarly, as per the findings from a database comprising three datasets of phase I and phase II trial, showed 57% of patients had an objective response with 10 months as the median duration of response [102]. Both larotrectinib and entrectinib have received tumor-agnostic approvals from the Food and Drug Administration (FDA).

### 5.5. Claudin 18 Agents

Claudins (CLDNs) represent one of the most recent targets in GC. Zolbetuximab represents the most mature of the efforts to target CLDN18.2. It is a mouse chimeric mAb with a human IgG1 constant region that specifically and with high affinity binds to CLDN18.2 through its first extracellular domain and causes ADCC and complement-dependent cytotoxicity mediated lysis of CLDN18.2 expressing GC cells [54]. SPOTLIGHT is a global, phase III, randomized trial conducted in patients with CLDN18.2-positive (defined as ≥75% of tumor cells showing moderate-to-strong membranous CLDN18 staining), HER2-negative, locally advanced unresectable or metastatic gastric or gastroesophageal junction adenocarcinoma, where patients were randomized to receive modified FOLFOX and zolbetuximab or mFOLFOX and placebo [103]. The results showed a significant reduction in the risk of disease progression or death with an HR of 0.75 [103]. GLOW is also a global phase III trial involving the randomized administration of zolbetuximab or placebo in combination with capecitabine and oxaliplatin (CAPOX) [104]. It showed significant prolongation of both PFS (median 8.21 versus 6.80 months, HR 0.687, *p* = 0.0007) and median OS 14.39 versus 12.16 months, HR 0.771, *p* = 0.0118) [104]. CMG 901 is an anti-claudin antibody-drug conjugate against CLDN18.2 with anticancer activity dependent on monomethyl auristatin E (MMAE)-mediated cytotoxicity [105]. In a phase I multicenter dose escalation trial in patients with CLDN18.2-positive gastric/GEJ cancer, the objective response rate was 75%, and the disease control rate (DCR) was 100% [105]. Similarly, SYSA1801 is also an MMAE antibody–drug conjugate targeting CLDN18.2, which in an open-label phase I study presented at the ASCO annual meeting this year shows ORR and DCR of 47.1% and 64.7% in patients with GC who were refractory or intolerant to standard therapies [106].

## 6. Immune Checkpoint-Targeted Therapies and Other Immunotherapies

Immunotherapy has rapidly evolved to be a pillar of oncologic treatment. Several immune checkpoints have been identified including programmed cell death 1 (PD-1), cytotoxic T-lymphocyte-associated protein 4 (CTLA-4), and lymphocyte activation gene 3 (LAG3).

The search for a reliable biomarker of immunotherapy response has been extensively studied, trials using either prospective or retrospective PD-L1 expression appear to have some discriminatory ability; however, there is no standard, likely due to the dynamic nature of these markers [107].

### 6.1. Immune Checkpoint Inhibitors in Metastatic/Advanced Gastric Cancer

Several studies have explored the role of PD-1 and PD-L1 inhibitors in advanced gastric cancer. One of the first FDA-approved PD-1 inhibitors was pembrolizumab, which was based on the phase II multi-cohort KEYNOTE-059 trial [108]. These enrolled patients in third-line or later advanced or metastatic gastric or gastroesophageal junction cancer with a PD-L1 CPS score ≥1 [108]. The objective response rate was 15.5% in patients with PD-L1-positive disease. Interestingly, in cohort 2 of the KEYNOTE-059 study, pembrolizumab plus chemotherapy (cisplatin plus 5-fluorouracil or capecitabine) demonstrated remarkable antitumor activity with an objective response rate of 60%. One of the first reported phase III trials was KEYNOTE-062, based on the promising results of KEYNOTE-059 trial, randomized 1:1:1 between pembrolizumab, pembrolizumab plus chemotherapy (cisplatin plus fluorouracil or capecitabine), or chemotherapy plus placebo [109]. This study enrolled 763 patients with advanced gastric or gastroesophageal cancer in 29 countries. The primary end point was OS in patients with PD-L1 CPS of 1 or greater. This was a negative study that demonstrated that pembrolizumab was not superior when compared with chemotherapy median OS 10.6 versus 11.1 months; hazard ratio HR, 0.91; 99.2% CI, 0.69–1.18). Although not statistically tested, there does seem to be a signal for CPS of 10 or greater. This further demonstrated that pembrolizumab was well tolerated and had fewer treatment-related adverse events when compared with chemotherapy. The phase III KEYNOTE-859 trial will attempt to demonstrate whether pembrolizumab in addition to the investigator’s choice of chemotherapy (fluorouracil and cisplatin or capecitabine and oxaliplatin) will be superior to chemotherapy alone [110]. The first readout of this trial will be presented at the European Society for Medical Oncology later this year, median OS was 12.9 month in the combination arm versus 11.5 months with placebo (HR 0.78, 95% CI 0.70–0.87, *p* < 0.0001). The median PFS was 6.9 months versus 5.6 months (HR 0.76, 95% CI 0.67–0.85, *p* < 0.0001).

In a mainly Asian population, the randomized phase II/III ATTRACTION-4 study enrolled 724 previously untreated HER2-negative, unresectable, advanced, or recurrent gastric or gastroesophageal junction cancer, regardless of PD-L1 expression, randomized to nivolumab plus chemotherapy (S-1 plus oxaliplatin or capecitabine plus oxaliplatin) versus chemotherapy plus placebo [111]. Despite an overall response rate of 65.8% (95% CI: 48.6–80.4), co-primary endpoints were PFS and OS. PFS in the nivolumab plus chemotherapy group was 10.45 months compared with 8.34 months in the placebo plus chemotherapy group (HR 0.68, CI 0.51–0.90; *p* = 0.0007). However, the median OS was 17.45 months in the nivolumab plus chemotherapy group and 17.15 months in the placebo plus the chemotherapy group (HR 0.90; 95% CI 0.75–1.08; *p* = 0.26). Again, this was largely a negative trial given the more important OS data.

The trial which led to the wide adoption of immunotherapy in advanced gastric cancer is Checkmate-649 [112], which is the first PD-1 inhibitor to demonstrate OS benefit. This was a three-arm phase III trial which enrolled 1581 patients randomized to nivolumab plus chemotherapy (capecitabine and oxaliplatin), nivolumab plus ipilimumab, a cytotoxic T-lymphocyte-associated protein 4 (CTLA-4 inhibitor), or chemotherapy alone. Primary endpoints were OS and PFS in patients with a PD-L1-combined positive score of five or more. Nivolumab plus chemotherapy resulted in significant improvements in OS (HR 0.71, 98% CI 0.59–0.86]; *p* < 0.0001) and PFS (HR 0.68, 98% CI 0.56–0.81]; *p* < 0.0001) when compared with chemotherapy alone in patients with a PD-L1 CPS of five or more (minimum follow-up 12.1 months).

More recently, presented at the American Association for Cancer Research 2023, another PD-1 inhibitor sintilimab showed improvements in OS in a purely Chinese population [113]. This study enrolled 650 patients with untreated, unresectable locally advanced or metastatic G/GEJ adenocarcinoma randomized to sintilimab in combination with chemotherapy (capecitabine and oxaliplatin) compared to placebo plus chemotherapy. They demonstrated improved OS with a 31.9% reduction in the risk of death (HR 0.681, 95% CI: 0.571–0.812, *p* < 0.0001) and median OS of 15.2 months versus 12.3 months in all randomized patients. Taken together, these large phase III trials clearly demonstrate that combining immune checkpoint inhibitors with chemotherapy is rapidly becoming the new standard first-line treatment for patients with advanced gastric or gastroesophageal cancer.

### 6.2. Immune Checkpoint Inhibitors in Locally Advanced Gastric Cancer

Early results from phase I and II trials in gastric cancer suggest that neoadjuvant checkpoint blockade is safe and increases pathologic complete response (pCR) rates [114].

The phase II study, known as NEONIPIGA (NCT04006262), examined the effectiveness of neoadjuvant immunotherapy using nivolumab and ipilimumab in patients with locally advanced resectable gastric or gastroesophageal junction (GEJ) adenocarcinoma with deficient DNA mismatch repair (dMMR) or high microsatellite instability (MSI H) [115]. Thirty-two patients were enrolled between October 2019 and June 2021 and 29 underwent surgery and 58.6% achieved pCR, suggesting its potential as a treatment approach for this patient population [115].

The Checkmate 577 trial established 1 year of adjuvant nivolumab as the standard of care for patients with esophageal cancer who previously received the CROSS protocol in neoadjuvant setting followed by surgical resection with residual pathologic disease [116] This study included both adenocarcinoma as well as squamous cell carcinoma, which contributed to 29% of patients. They demonstrated disease-free survival was significantly longer in patients who had adjuvant nivolumab versus placebo, with a median DFS of 22.4 versus 11 months (HR 0.69, 96.4% CI 0.56–0.86, *p* < 0.001). Distant metastasis-free survival was also improved from a median of 28.3 to 17.6 months (HR 0.74, 95% CI 0.60–0.92).

Given the exciting results of Checkmate 577, multiple phase III trials will examine whether the addition of immune checkpoint blockade to perioperative chemotherapy will provide similar benefits in patients with gastric and gastroesophageal cancers. One highly anticipated trial is the largely European phase IIb DANTE study, which will explore whether the addition of atezolizumab to perioperative FLOT will improve its primary outcome of DFS/PFS [117]. The interim results presented at the American Society for Clinical Oncology showed that the trial enrolled 295 patients with resectable adenocarcinoma of the stomach and GEJ (≥cT2 and/or N+). The authors reported similar tolerability, but more pathological regression favoring atezolizumab, with pT0 rates of 23% versus 15%, pN0 rates of 68% versus 54% in the atezolizumab and FLOT arms, respectively.

Another highly anticipated trial is the international MATTERHORN trial, which is recruiting at 175 institutions, will study the combination of durvalumab with FLOT (fluorouracil, leucovorin, oxaliplatin, and docetaxel) versus FLOT alone [118]. This is a large phase III randomized trial evaluating the efficacy of neoadjuvant–adjuvant durvalumab in patients with resectable gastric and gastroesophageal cancers. Like DANTE, the primary endpoint is EFS, and the secondary endpoint will explore OS and pCR. There was a press release in June of 2023 which suggested the immunotherapy combination improved pathologic complete response; however, this is not yet peer reviewed. Finally, in a large Asian population, ATTRACTION-05 aimed to examine the combination of nivolumab with adjuvant S-1 or CAPOX chemotherapy will improve the primary endpoint of RFS [119]. This is a randomized phase III trial in the adjuvant setting for patients with pathologic stage III G/GEJ cancer with treatment continued for up to 1 year. Disappointingly, the first results reported at the 2023 American Society for Clinical Oncology of the primary efficacy endpoint of RFS was not met (HR 0.90, 95.72% CI, 0.69–1.18, *p* = 0.4363), with the 3-year RFS rates of 68.4% in the combination arm and 65.3% in the chemotherapy alone arm. Keynote 585 is similarly a phase III study evaluating perioperative chemotherapy with or without Pembrolizumab in patients with GC [120]. At a pre-specified interim analysis, while the study met its primary endpoint of improved pCR in the treatment arm, it did not meet significance for improvement in event-free survival [EFS] [121]. Ongoing studies will be needed to better select which patients will ultimately benefit from combination treatment.

### 6.3. Immune Checkpoint Combined with Target Therapies

While the addition of the anti-programmed death 1 (PD-1) antibody pembrolizumab to chemotherapy does not notably enhance efficacy in advanced HER2-negative GC [110], there are both preclinical [122] and clinical [123] reasons to consider adding pembrolizumab for HER2-positive disease. The phase III study Keynote 811 tested the combination of pembrolizumab plus trastuzumab and chemotherapy for unresectable or metastatic, HER2-positive gastric or gastroesophageal junction adenocarcinoma [124]. The first interim analysis demonstrated a significant reduction in tumor size, induction of complete responses in some participants, and a marked improvement in the objective response rate due to the addition of pembrolizumab 74.4% (66.2–81.6) in the Pembrolizumab group versus 51.9% (43.0–60.7) in the placebo group. However, based on a pre-specified subgroup analysis by PD-L1 expression, the improvement in PFS observed in the intention-to-treat (ITT) population was limited to patients whose tumors were PD-L1 positive (CPS ≥ 1). In the study, more than 80% of patients had tumors that were PD-L1 positive. A trend toward improvement in OS, the trial’s other primary endpoint, was observed in the ITT population for patients who received the pembrolizumab combination versus placebo in combination with trastuzumab and chemotherapy; however, these results did not meet statistical significance per the pre-specified statistical analysis plan.

Another study combining anti-PD-1, and target therapy is the ongoing clinical trial, HERIZON-GEA-01 [125]. This is a global phase III study aiming to evaluate the effectiveness and safety of zanidatamab, a novel bispecific HER2-targeting monoclonal antibody, in combination with chemotherapy with or without tislelizumab, an anti-PD-1 antibody, as a first-line treatment for advanced/metastatic HER2+ gastroesophageal adenocarcinomas [125].

### 6.4. Immune Checkpoint Inhibitors in Microsatellite Instability-High (MSI-H) Gastric Cancer

MSI is common in gastric cancer, affecting about 19% of patients in all stages [126]. Multiple studies have shown that patients with GEJ and gastric cancers with an MSI-H phenotype have a better prognosis regardless of disease stage [127]. Updated results from the Checkmate 649 trial show that in the MSI-H subset, patients who received nivolumab and chemotherapy had an OS of 38.7 months as compared to 12.3 months with chemotherapy alone [128]. MSI status has been shown to be significant for resectable gastroesophageal malignancies as well. The INFINITY trial is a multicenter, single-arm, multi-cohort, phase II trial of tremelimumab and durvalumab as neoadjuvant treatment or definitive treatment in resectable gastroesophageal junction adenocarcinoma and gastric adenocarcinoma with MSI-H disease [129]. These patients received a 3-month course consisting of a single high dose of tremelimumab 300 mg and three cycles of durvalumab 1500 mg given every 4 weeks (T300/D), followed by surgery. The results show that 60% of patients (9/15 patients) achieved pCR, and 80% achieved major pathologic response defined as <10% viable cells [129].

Figure 4 will present a concise summary of the latest novel targets discovered in the context of gastric cancer and Table 1 and Table 2 will provide a comprehensive description of the main trial studies, encompassing both target therapies and checkpoint inhibitors.

## 7. Conclusions

In summary, our review underscores the critical molecular pathways driving gastric cancer progression, including the MAPK, PI3K/AKT/mTOR, HGF/c-MET, and Wnt/β-catenin pathways. While these pathways hold promise as targets for therapeutic intervention, it is important to acknowledge the ongoing challenges in developing effective treatments for them.

Furthermore, our analysis highlights specific mechanisms and pathways for which targeted treatments are currently available. In cases of HER2-positive disease, HER2-directed agents have become the standard therapy, and promising newcomers like zanidatamab and trastuzumab deruxtecan show potential treatments in the first-line setting.

Immunotherapy agents have gained prominence in neoadjuvant and adjuvant treatment strategies in GC. However, it remains essential to pinpoint which patients truly derive the greatest benefit from these therapies. Notably, the disappointing event-free survival outcomes observed in the unselected patient population of the KEYNOTE 585 study in the neoadjuvant setting underscore this need for precision. In this context, we must intensify our exploration of biomarkers, including CPS, in localized disease. In metastatic disease, immune checkpoint inhibitors showed a potential benefit particularly in patients with a CPS ≥ 5.

The addition of the anti-PD-1 agent to chemotherapy and trastuzumab, for HER2-positive metastatic gastroesophageal adenocarcinoma in the phase III Keynote 811 study resulted in a significant improvement of tumor response rate. However, the enhanced progression-free survival was primarily observed in patients with PD-L1 positive, and OS did not meet statistical significance per the pre-specified statistical analysis plan. This study highlights that selection by biomarker, in this case PD-L1, appears to be fundamental to define which patient will benefit from the addition of immunotherapy.

Emerging agents, such as FGFR2b and Claudin 18.2 directed therapies, offer an additional option for HER2-negative metastatic disease. In later lines of therapy, VEGF inhibitors continue to play a significant role. As more data emerge, gastric cancer management will increasingly target its underlying molecular mechanisms.

## Figures and Tables

**Figure 1 cancers-15-05075-f001:**
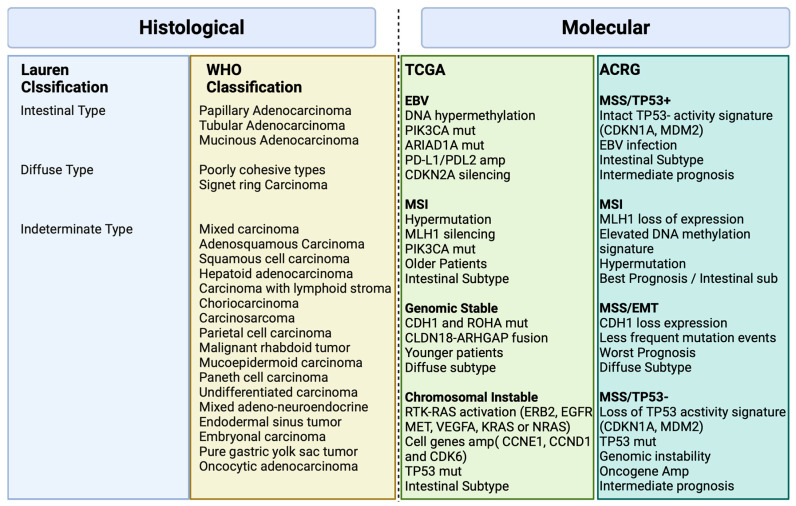
Histological and Molecular Classification in Gastric Cancer Biorender agreement number: JO25V7ORTC.

**Figure 2 cancers-15-05075-f002:**
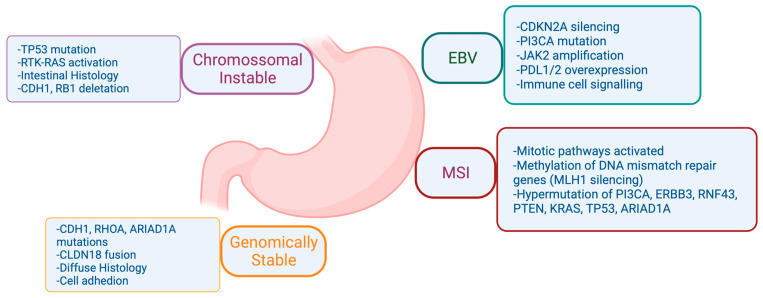
Comprehensive Molecular Characterization of Gastric Adenocarcinoma according to The Cancer Genome Atlas (TCGA). Biorender agreement number CU25Y0RJS4.

**Figure 3 cancers-15-05075-f003:**
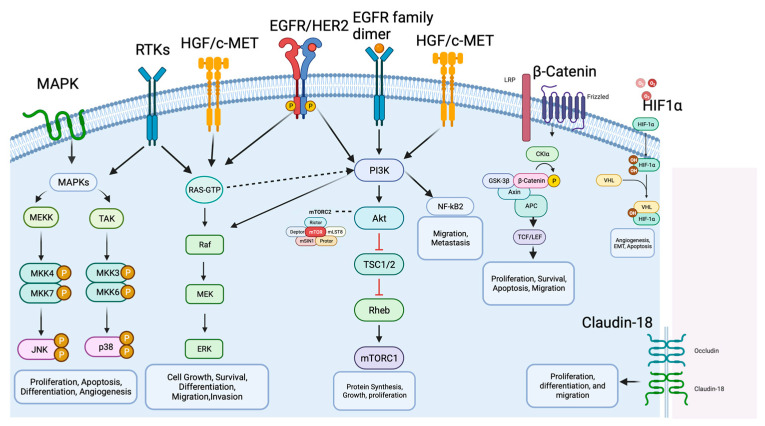
Molecular mechanisms and signaling pathways in gastric cancer. The figure illustrates the main molecular pathways and key factors involved in GC development and progression, including MAPK, HER2, PI3K/AKT/mTOR, HGF/c-Met, p53, Wnt/β-catenin, NF-κB and PD-1/PD-L1 and CTL4. The arrows represent activation signals and there are no difference between the arrows type (dashed versus solid). Biorender agreement number: VO25V7OL5G.

**Figure 4 cancers-15-05075-f004:**
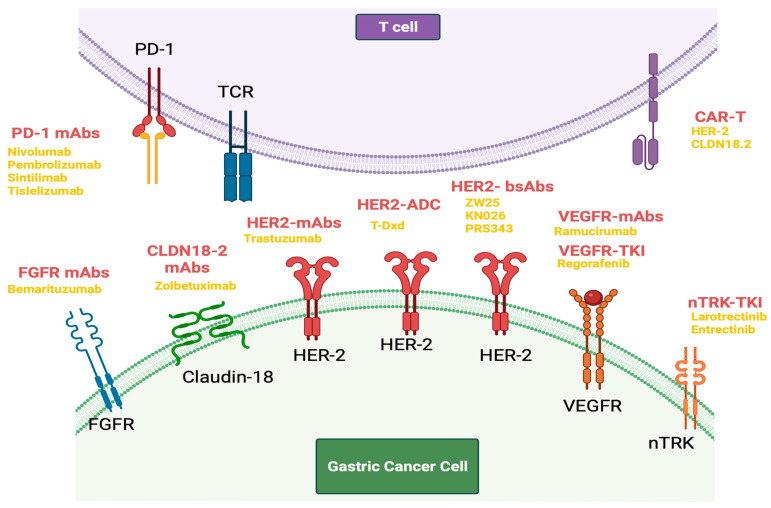
Immunotherapy and target therapy in metastatic gastric cancer. This figure provides guidance for currently available immunotherapy and targeted therapy based on different biomarkers. Biorender agreement number: WK25V7O0M0.

**Table 1 cancers-15-05075-t001:** Targeted therapies in metastatic gastric cancer.

Molecular Alteration	Trial	Phase	Patient Population	Treatment	Median PFS (Months)	Median OS (Months)
**HER 2** **Amplification**	ToGA	III	Treatment-naive	Capecitabine or 5FU + Ciplastin	5.5	11.1
				Capecitabine/5FU + Cisplatin + Trastuzumab	6.7	13.8
	DESTINY GASTRIC 01	II	Pretreated (3rd line or later)	Trastuzumab deruxtecan	5.6	12.5
				Chemotherapy (Irinotecan or Paclitaxel)	3.5	8.4
	DESTINY-GASTRIC 02	II	Pretreated (2nd Line)	Trastuzumab deruxtecan	4.1	7.9
	(NCT03929666) ZW-25	II	Treatment-naive	ZW25 + Chemotherapy	12.5	NR
**FGFR** **mutations**	FIGHT	II	Treatment-naive	Bemarituzumab + mFOLFOX6	9.5	NR
				Placebo + mFOLFOX6	7.4	12.9
**CLDN18.2**	SPOTLIGHT	III	Treatment-naive	Zolbetuximab mFOLFOX6	10.6	18.2
				Placebo + mFOLFOX6	8.6	15.5
**VEGF/** **VEGFR**	RAINBOW	III	Pretreated (2nd Line)	Paclitaxel	2.86	7.4
				Paclitaxel + Ramucirumab	4.4	9.6
	REGARD	III	Pretreated (2nd Line)	Ramucirumab	2.1	5.2
				Placebo	1.3	3.8
	INTEGRATE	II	Pretreated	Regorafenib	2.6	5.8
				Placebo	0.9	3.4

PFS, progression-free survival; not reached (NR), not reported; OS, overall survival.

**Table 2 cancers-15-05075-t002:** Checkpoint inhibitors therapies in metastatic gastric cancer.

Molecular Alteration	Trial	Phase	Patient Population	Treatment	Median PFS (Months)	Median OS (Months)
**PD-1**	KEYNOTE-062	III	Treatment-naïve G/GEJ	Pembrolizumab Pembrolizumab/chemotherapy Chemotherapy	6.2 6.9 6.4	10.6 12.5 11.1
	ATTRACTION-04	II/III	Treatment-naïve G/GEJ	Nivolumab/chemotherapy Chemotherapy	10.5 8.3	17.5 17.2
	CHECKMATE-649	III	Treatment-naïve G/GEJ	Nivolumab/chemotherapy Nivolumab/ipilimumab Chemotherapy	7.7 NA 6	14.4 NA 11.1
	ORIENT-16	III	Treatment-naïve G/GEJ	Sintilimab/chemotherapy Chemotherapy	NA	15.2 12.3
**MSI-H**	KEYNOTE-158	II	Non-colorectal, MSI-H, after fist-line	Pembrolizumab	4.1	23.5
**PD-1** **HER 2**	KEYNOTE-811	III	Treatment-naïve G/GEJ HER2+	Pembrolizumab/Trastuzumab/Chemotherapy Trastuzumab/Chemotherapy	To be released	Negative in ITT

PFS, progression-free survival; NR, not reported; OS, overall survival.

## Data Availability

Not applicable

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
