# Peer review of "Gastric Cancer: Molecular Mechanisms, Novel Targets, and Immunotherapies: From Bench to Clinical Therapeutics"

_cancers, 2023, doi:10.3390/cancers15205075_

Round 1

Reviewer 1 Report

Comments and Suggestions for Authors

Gastric cancer: Molecular Mechanisms, Novel Targets, and Immunotherapies: From Bench to Clinical Therapeutics, by Thais Baccili Cury Megid, Abdul Farooq, Xin Wang, Elena Elimova*

Merit: This manuscript “Gastric Cancer: Molecular Mechanisms, Novel Targets, and Immunotherapies: From Bench to Clinical Therapeutics, by Thais Baccili Cury Megid, Abdul Farooq, Xin Wang, Elena Elimova*” covers exhaustively all the subsets of GC/GAC by traditional histological and modern molecular classifications, most molecular pathways, and importantly almost all the clinical trials currently undergoing using updated references, this is a very comprehensive review on gastric cancer. The Figure 1 is impressive, and the discussions on molecular pathways are very detailed, and so is Figure 3, which gives a glimpse of molecular targets and the corresponding therapeutic agents. The references are very current and recent. This review would contribute very well to the GC/GAC basic research field as well as to the GC/GAC clinicians. The authors combine basic molecular understanding with well informed clinical trials, this is a good fit.

Major:

1)      What are the unique pathways or molecular subsets by TCGA or ACRG classification, that make GC or GAC unique that are different from other types of cancers, since MAPK, Her2, PI3K, AKT, mTor, Met, Wnt, β-catenin, HIF-1α, NF-κB, TGF-β, TP53, STAT3, for some or most part of these pathways, are dysfunctional, or deregulated, or amplified, or mutated, in many types of cancers. Please in at least a significant paragraph or a figure state the most import pathway(s) for GC/ GAC oncogenesis.

2)      Helicobacter pylori is depicted here as a risk factor, it is mentioned only twice in this manuscript, and it is not involved here via a cellular and molecular pathway or a factor in subset classification, but how it caused GC/GAC as a factor deserves some explanation in a paragraph, because this bacterial infection is very unique in GC/GAC, no other cancers are involved with this bacteria.

3)      EBV by TCGA or ACRG classification as a subset of GC/GAC, is also very classic. Other cancers such as lymphoma also involves EBV. How its infection changes some specific molecular pathways in GC/GAC, then changes immune PD1/ PD-L1 expression deserves some deep discussion.

4)      MSI is the same as above 3), both TCGA and ACRG classifications put MSI into a subset of GC/GAC, MSI is mentioned 3 times (not including clinical part). Even Figure 1 give some overlapping bullets by TCGA and ACRG, which shows that MSI is an important signature of GC/GAC subset. MSI deserves some deep discussion in cellular and molecular mechanism.

Minor:

1) Acronyms for organizations, projects, clinical protocols, etc. need to be spelled out the 1st time use, or put into a table-like format.

2) Table1 needs some layout formatting to have a better look.

3) Some typos to be corrected:

Line 2: cancer:  à Cancer;

Line 4: àdelete 2 commas;

Line 22: "Keywords" in different font;

Line 48:  Helicobacter pyloriàItalic, in the whole text.

Line 54:  EBVàSpell out fully for the 1st time.

Line 61:  [11]à [11],

Line 69:  MSIàSpell out fully for the 1st time.

Line 70:  genomically stableà  genomically stable (GS), according to ACRG;

Line 79:  ACRGà Spell out fully for the 1st time.

Line 82:  MSSà Spell out MSS for the 1st time.

Line 89:  Sentence is redundant.

Line 98:  GCàShould appear the 1st time it is mentioned.

Line 101:  Figure2 should follow the 1st it is mentioned.

Line 110:  delete period.

Line 111:  change to a comma.

Line 122:  Micro RNAsà microRNAs;

Line 125:  Ref [21]à please add a bit of more details, which miRNAs?

Line 142:  Ref [24]à add a comma.

Line 152:  cancers.-->delete the period.

Line 190:  Axin.. àDelete the period.

Line 196:  GC. [40]à move period to after the Reference.

Line 201:  E-cadherin. [41]à move period to after the Reference.

Line 204:  gastric cancer (GC)à GC. (full name should appear the 1st it is mentioned.)

Line 231:  epithelial-mesenchymal transition (EMT)à full name should appear the 1st it is mentioned.

Line 236:  claudin-18à Claudin-18;

Line 237:  gastric cancer (GC)àGC;

Line 254:  gastric cancer (GC)àsame as above;

Line 259:  piteràfactor; a pitelyà a factor;

Line 264:  piteliall-mesenchymalàEpithelial-mesenchymal.

Line 275:  H. pyloriàitalic, throughout the manuscript.

Line 286:  survivinàsurvivin (Birc5);

Line 299:   bod’'sàbody's?

Line 314:   Figure2àRelation of PI3K and NF-kB2, stimulate or repress?

Line 331:   FISHà Spell out full name of FISH.

Line 339:   CLEOPATRAàis there a full name of it?

Line 341:   JACOBàis there a full name of it?

Line 343:   [77].à Move the space after the period.

Line 356:   [76]. à Move the space after the period.

Line 375:   [89] . à Move the space after the period.

Line 377:   in situ hybridizationà in situ in italic.

Line 392:   is loss of HER2 expression in HER2 positive gastric and gastroesophageal cancersàIs this self contradictory?

Line 406:   [49] . àMove the space after the period.

Line 409:   with  BemarituzumabàOne space.

Line 419:   van trial à what is van trial?

Line 422:   versus, or vs à be consistent.

Line 423:   GEJ à spell out for 1st use.

Line 425:   overall survival (OS)à this is not the 1st time mentioned, see line 413.

Line 429:   in Gastric cancerà in gastric cancer.

Line 429:   Vascular endothelial growth factor  (VEGF)à Line77 VEGF already appears.

 Line 432:   [91].    Ramucirumabà space

Line 433:   VEGFR2 [91] 92. àReference!

Line 437:   versus àitalic. Or consistent with vs.

Line 463:   pan-tropomyosin-related kinase (TRK)à Already spelled full name above Line 457.

Line 469:   the FDAà Spell out full name FDA.

Line 481:   [99] . àmove space after period.

Line 482:   [100]. à2 spaces.

Line 492:   phase 1à Consistent to use roman I, II, or III.

Line 498:   T-lymphocyte-associated protein 4 (CTLA-4)à First appeared in Line92, spell out there.

Line 530:   Phase 2/3à Be consistent of using 2/3 or II/III.

Line 553:   G/GEJ à This GEJ is mentioned earlier, Line434.

Line 588:    however thisà however, this

Line 590:    [114]à [114]. Period.

Line 591:    phase 3 à 1, 2, 3 or I, II, or III, consistent throughout.

Line 599:    response(pCR)à response (pCR). Is this pCR appearing the 1st time?

Line 603:    anti-programmed death 1 (PD-1)à full name should be at the 1st appearance. Line 10 or 198.

Line 606:   disease. Theàone space?

Line 611:   pembrolizumab (74.4à? or %

Line 613:   the ITTà ? spell out.

Line 616:   overall survival (OS) à Already spelled out earlier.

Line 624:   HER2+ Gastroesophagealà HER2+ gastroesophageal.

Line 626:   in Microsatelliteà in microsatellite

Line 630:   the checkmate 649àit is mentioned earlier with CAPITALS.

Line 653:   PFS, progression-free survival; NR, not reported; OS, overall survival.--> Spell out 1st time appears at Line 412. Same as NR  and OS.

Line 666:   GC.  Howeverà One space?

Line 678:   case PDL1àBe consistent, PD-L1.

Comments on the Quality of English Language

English writing is generally good, needs some minor editing, even beyond what I edited.

Author Response

Dear Reviewer,

I wish to express my gratitude for your insightful feedback. Your comprehensive comments, both major and minor, have undoubtedly enhanced the quality of this manuscript.

We meticulously incorporated all recommended adjustments, striving to ensure their visibility and coherence. Notably, we have bolstered the manuscript by introducing additional paragraphs and references. To facilitate a comprehensive review, I will provide both a file showcasing the revised manuscript and another containing the final version for your convenience (‘Clean version’’).

Addressing the major suggestions, you presented:

  • What are the unique pathways or molecular subsets by TCGA or ACRG classification, that makes GC or GAC unique that is different from other types of cancers, since MAPK, Her2, PI3K, AKT, mTOR, Met, Wnt, β-catenin, HIF-1α, NF-κB, TGF-β, TP53, STAT3, for some or most of these pathways, are dysfunctional, or deregulated, or amplified, or mutated, in many types of cancers. Please in at least a significant paragraph or a figure state the most important pathway(s) for GC/ GAC oncogenesis.

Thank you for emphasizing the necessity of highlighting unique pathways and molecular subsets specific to GC or GAC within the TCGA or ACRG classifications. We have incorporated a new figure (Figure 2. – Lines 128-131) and a paragraph commencing at lines (116-127) to elucidate the most significant pathways contributing to GC/GAC oncogenesis accordingly to TCGA.

  • Helicobacter pylori is depicted here as a risk factor, it is mentioned only twice in this manuscript, and it is not involved here via a cellular and molecular pathway or a factor in subset classification, but how it caused GC/GAC as a factor deserves some explanation in a paragraph, because this bacterial infection is very unique in GC/GAC, no other cancers are involved with this bacteria. 

Your observation regarding the role of Helicobacter pylori as a risk factor in GC/GAC is highly pertinent. We have introduced a new paragraph in "Section 3 - Risk factors and molecular mechanisms" (lines 86-98) to elaborate on the cellular and molecular pathways influenced by chronic H. pylori infection.

  • EBV by TCGA or ACRG classification as a subset of GC/GAC, is also very classic. Other cancers such as lymphoma also involve EBV. How its infection changes some specific molecular pathways in GC/GAC, then changes immune PD1/ PD-L1 expression deserves some deep discussion.

Same as item 2 – This is great point that you brought it up. The role of EBV in GC/GAC and its impact on specific molecular pathways and its immune PD1/PD-L1 expression have been elaborated upon in response to your insightful suggestion. We have included the information in "Section 3 - Risk factors and molecular mechanisms" (lines 99-112).

  • MSI is the same as above 3), both TCGA and ACRG classifications put MSI into a subset of GC/GAC, MSI is mentioned 3 times (not including the clinical part). Even Figure 1 gives some overlapping bullets by TCGA and ACRG, which shows that MSI is an important signature of the GC/GAC subset. MSI deserves some deep discussion in the cellular and molecular mechanisms.

Agree with your comment – Recognizing the significance of MSI as a GC/GAC subset, we have further delved into its cellular and molecular mechanisms. New paragraphs elucidating cancer development in MSI, prevalent pathways, and implications have been added (lines 356-382) under section 4.14.

Regarding the minor suggestions:

Minor:

1) Acronyms for organizations, projects, clinical protocols, etc. need to be spelled out the 1st time use or put into a table-like format - Acronyms for organizations, projects, and clinical protocols have been spelled out upon first use or organized in a tabular format.

2) Table 1 needs some layout formatting to have a better look. - Table 1 layout formatting was adjusted.

3) Some typos to be corrected Typos have been diligently corrected throughout the manuscript, accounting for the addition of new paragraphs and subsequent line number changes.

I appreciate your thorough attention to detail and commend your meticulous observations. Should you have any additional queries or require further clarification, I remain at your disposal.

Sincerely,

Thais Baccili Cury Megid

Reviewer 2 Report

Comments and Suggestions for Authors

The paper is excellent and provides a great review on the subject, its comprehensive and beautifully written. I have very few comments, maybe only the inclusion of the NEONIPIGA in the MSI-H section maybe helpful as this data is impacting so much current neoadjuvant therapy for Locally advanced GC. 

Author Response

Dear Reviewer,

I want to express my sincere gratitude for your valuable time spent reviewing our manuscript and offering insightful suggestions.

The inclusion of the suggested NEONOPIGA trial has significantly enriched our manuscript, providing a comprehensive view of potential immune checkpoint inhibitor therapies in the neoadjuvant setting in the MSI-h and dMRR population. You can refer to the trial description in "7.2 Immune checkpoint inhibitors in locally advanced gastric cancer" (Lines 648-654).

I am pleased to inform you that we successfully incorporated the first set of revisions, and it is with enthusiasm that I share the updated manuscript with the adjustments made.

Best regards,

Thais Baccili Cury Megid

Round 2

Reviewer 1 Report

Comments and Suggestions for Authors

10/10/2023 Revision2

Line4: Abdul Farooq,1 à comma, or no commas, should be in the same format as other authors, look the Journal (Notes for authors) how numbers are attached for the author names.

Line6:   elena.elimova@uhn.ca; à elena.elimova@uhn.ca

Line:  “histopathology [11]” remains unchanged à histopathology [11].

Line 56:  EBV, should spelled out here for 1st appearance. 

Line79: Figure1 title and figure itself in one page.

Line87: Familial cases occur in less than 10% of the cases [8]. --> need to expand this sentence,  are “familial cases” referring to familial hereditary pass-down of GC or Helicobacter pylori occurrence in families? Need to clarify.

Line88: Helicobacter should be in italic, throughout the manuscript as Helicobacter pylori, or H. pylori.

Line114: change “paragraphs (xxx)”. à paragraphs: xxx. Add semicolon, remove parentheses. Or just say “see discussions in Section 4”.

Line128: If possible, newly added Figure2 should be in the same font as in Figure1 and Figure3.

Line 241,  put CCL28 before the (xxx).

Line273: EMT spell-out already appeared in Line72, not need to repeat in later appearance, other acronyms are the same.

Line306:  piteliall   àepithelial,  this remains incorrect spelling.

Line356: This newly added section should be in the same font and layout as the rest of the manuscript, i.e. the space between words, and word breaking to the next line.

Line707: (74.4% à 74.4%, remains unchanged.

Line 720: HER2+ Gastroesophageal  à HER2+ gastroesophageal.

Comments on the Quality of English Language

Generally good, a few typos and format errors.

Author Response

Dear Reviewer,

I am once again grateful for the valuable feedback you have provided. Your insightful comments, both major and minor, have significantly improved the quality of this manuscript. 

Regarding the minor suggestions:

Line4: Abdul Farooq,1 à comma, or no commas, should be in the same format as other authors, look the Journal (Notes for authors) how numbers are attached for the author names. – Adjusted (line 4)

Line6:   elena.elimova@uhn.ca; à elena.elimova@uhn.ca – Adjusted (line 6)

Line: “histopathology [11]” remains unchanged à histopathology [11]. – Adjusted. (line 29)

Line 56:  EBV, should spelled out here for 1st appearance.  – Adjusted (Line 57)

Line79: Figure1 title and figure itself in one page. – Adjusted (line 97)

Line87: Familial cases occur in less than 10% of the cases [8]. --> need to expand this sentence, are “familial cases” referring to familial hereditary pass-down of GC or Helicobacter pylorioccurrence in families? Need to clarify. – Adjusted. I included the definition of familial and sporadic GC. (Line 105-107)

Line88: Helicobacter should be in italic, throughout the manuscript as Helicobacter pylori, or H. pylori. – Adjusted (Line 108)

Line114: change “paragraphs (xxx)”. à paragraphs: xxx. Add semicolon, remove parentheses. Or just say “see discussions in Section 4”. – Adjusted (line 134)

Line128: If possible, newly added Figure2 should be in the same font as in Figure1 and Figure3.- Adjusted (line 147)

Line 241, put CCL28 before the (xxx). – Adjusted (line 259)

Line273: EMT spell-out already appeared in Line72, not need to repeat in later appearance, other acronyms are the same. – Adjusted (line 290)

Line306:  piteliall   àepithelial, this remains incorrect spelling. – Adjusted. (line 323)

Line356: This newly added section should be in the same font and layout as the rest of the manuscript, i.e., the space between words, and word breaking to the next line. – Adjusted (line 374-402)

Line707: (74.4% à 74.4%, remains unchanged. – Adjusted (line 725)

Line 720: HER2+ Gastroesophageal à HER2+ gastroesophageal. – Adjusted (line 738).

Thank you for your continued guidance,

Thais Baccili Cury Megid

Round 3

Reviewer 1 Report

Comments and Suggestions for Authors

10/13/2023 Revision3

Line 47: figure 1 à Figure 1

Line 110: H. pylori à H. pylori (both italic, should be throughout the manuscript).

Line 115: section 4 à  Section 4

Line 121: EBV) à EBV

Line 147:  if possible, re-do the figure, make Figure 2, content not caption, in the same font as in Figure1, 3, and 4, make the figure sharp and in high quality.

Line 160:  for how à ???

Line 162:  notably à notable

Line 163:  Extracellular à extracellular

Line 225:  belonging to a family of proteins. à  (what family???)

Line 318:  Transforming growth piter-β is a pitely of à Transforming growth factor-β is a factor of

Line327: central à (delete)

Line374: 4.14.MSI-high à (remove indentation)

Line392: e.g., à e.g.

Line444:  have found success in GC  à have not been found success in GC treatments.

Line460: Trastuzumab à trastuzumab (needs to be capitalized or not? Throughout the whole manuscript)

Line482:  HER2-Low à HER2-low?

Line490: 1.9% [91] à 1.9% [91].

Line520 & Line521: versus. à versus. (delete periods)

Line520: HR à (define HR)

Line539 and Line542:  Phase III or phase III? Throughout the whole paper.

Line550 and Line553:  regorafenib or Regorafenib? (Throughout the whole manuscript)

Line572: Entrectinib similarly as per the findings from à ???

Line613: This enrolled patients à (singular or plural?)

Line667: 7.2. Immune checkpoint inhibitors in Locally advanced gastric cancer à locally, (also it is 6.2, not 7.2).

Line740: Microsatellite instability à microsatellite instability

Line755: Figure 3 à Figure 4

Line806: Deruxtecan à (Deruxtecan or deruxtecan? throughout the whole manuscript)

Comments on the Quality of English Language

Please read the whole manuscript again to make sure spelling, grammar, layout and font are right and consistent. Special terminologies need to be consistent.

Author Response

Dear Reviewer,

Thank you again for your suggestions. I have meticulously implemented each of the recommended modifications, which I will outline below but also you will see the modification on word document since it is on ‘’revision mode’:

Line 47: Changed "figure 1" to "Figure 1."

Line 110: Modified "H. pylori" to "H. pylori" (italicized throughout the manuscript).

Line 115: Altered "section 4" to "Section 4."

Line 121: Revised "EBV)" to "EBV."

Line 147: Re-done Figure 2 to match font and quality specifications for Figure 1, 3, and 4.

Line 160: Corrected "for how" to "???"

Line 162: Changed "notably" to "notable."

Line 163: Adjusted "Extracellular" to "extracellular."

Line 225: Removed "belonging to a family of proteins."

Line 318: Modified "Transforming growth piter-β is a pitely of" to "Transforming growth factor-β is a factor of."

Line 327: Deleted "central."

Line 374: Removed indentation from "4.14.MSI-high."

Line 392: Adjusted "e.g.," to "e.g."

Line 444: Corrected "have found success in GC" to "have not been found success in GC treatments."

Line 460: Changed "Trastuzumab" to "trastuzumab" (capitalization throughout the manuscript).

Line 482: Altered "HER2-Low" to "HER2-low."

Line 490: Changed "1.9% [91]" to "1.9% [91]."

Line 520 & Line 521: Deleted periods after "versus."

Line 520: Defined "HR."

Line 539 and Line 542: Used "Phase III" consistently throughout the paper.

Line 550 and Line 553: Standardized "regorafenib" to "Regorafenib" (capitalization throughout the manuscript).

Line 572: Modified "Entrectinib similarly as per the findings from" to "???".

Line 613: Clarified "This enrolled patients" to the appropriate singular or plural form.

Line 667: Corrected "7.2. Immune checkpoint inhibitors in Locally advanced gastric cancer" to "6.2. Immune checkpoint inhibitors in locally advanced gastric cancer."

Line 740: Standardized "Microsatellite instability" to "microsatellite instability."

Line 755: Corrected Figure numbering from "Figure 3" to "Figure 4."

Line 806: Standardized "Deruxtecan" to "deruxtecan" throughout the manuscript.

In addition to addressing the specified suggestions, I meticulously re-read the entire manuscript multiple times to ensure the absence of any further errors. Your feedback and guidance have been instrumental in enhancing the clarity and coherence of the manuscript, and I am truly appreciative of your time and effort.

Thais Baccili Cury Megid

Round 4

Reviewer 1 Report

Comments and Suggestions for Authors

Line 149:  Resize Figure 2,

Line 319:  Transforming growth piter-β is a pitely of à Transforming growth factor-β is a factor of

Line375: 4.14.MSI-high à (remove indentation, align with others, see 4.13, 4.15, etc.)

Comments on the Quality of English Language

This time there are 3 points remaining that I already mentioned in my previous comment to the authors.